# Solving Graph-based Public Good Games with Tree Search and Imitation Learning

**Victor-Alexandru Darvariu**[1,2], **Stephen Hailes**[1], **Mirco Musolesi**[1,2,3]
[1]University College London    [2]The Alan Turing Institute    [3]University of Bologna
{v.darvariu, s.hailes, m.musolesi}@cs.ucl.ac.uk

## Abstract

Public goods games represent insightful settings for studying incentives for individual agents to make contributions that, while costly for each of them, benefit the wider society. In this work, we adopt the perspective of a central planner with a global view of a network of self-interested agents and the goal of maximizing some desired property in the context of a best-shot public goods game. Existing algorithms for this known NP-complete problem find solutions that are sub-optimal and cannot optimize for criteria other than social welfare.

In order to efficiently solve public goods games, our proposed method directly exploits the correspondence between equilibria and the Maximal Independent Set (mIS) structural property of graphs. In particular, we define a Markov Decision Process which incrementally generates an mIS, and adopt a planning method to search for equilibria, outperforming existing methods. Furthermore, we devise a graph imitation learning technique that uses demonstrations of the search to obtain a graph neural network parametrized policy which quickly generalizes to unseen game instances. Our evaluation results show that this policy is able to reach 99.5% of the performance of the planning method while being three orders of magnitude faster to evaluate on the largest graphs tested. The methods presented in this work can be applied to a large class of public goods games of potentially high societal impact and more broadly to other graph combinatorial optimization problems.

## 1 Introduction

In a *public goods game* (PGG), individuals can choose to invest in an expensive good (paying a cost), with benefits being shared by wider society [37]. It is a form of $n$-party social dilemma that has been used to study the tension between decisions that benefit the individual and the common good [36]. Aspects characteristic to public goods are observed in many important societal problems such as meeting climate change targets [32, 41], the dynamics of research and innovation [29], the design of effective vaccination programs [19], and, more generally, situations in which contributions are non-excludable. The analysis of this class of games is related to ongoing efforts to study cooperation in multi-agent systems as a means of driving progress on societal challenges [13].

The *best-shot* PGG is a variant in which investment decisions are binary and agents benefit if either they or a neighbor own the good [26]. Since patterns of connections along social and geographical dimensions in networks are known to shape individual decision-making [9, 23], a natural restriction is to limit the impact of contributions to an agent's neighbors. Graph-based best-shot public goods games exhibit multiple pure-strategy Nash equilibria (PSNE) [16]. Given this multiplicity, a natural question that arises is how to compute equilibria that satisfy some properties, a task known to be NP-complete in general for multiplayer games [22, 12]. Examples of desirable equilibria are those that maximize the social welfare (total utility) of agents or those with a high degree of fairness in terms of contributions. For graph-based best-shot PGGs, it has been shown that each equilibrium

35th Conference on Neural Information Processing Systems (NeurIPS 2021).

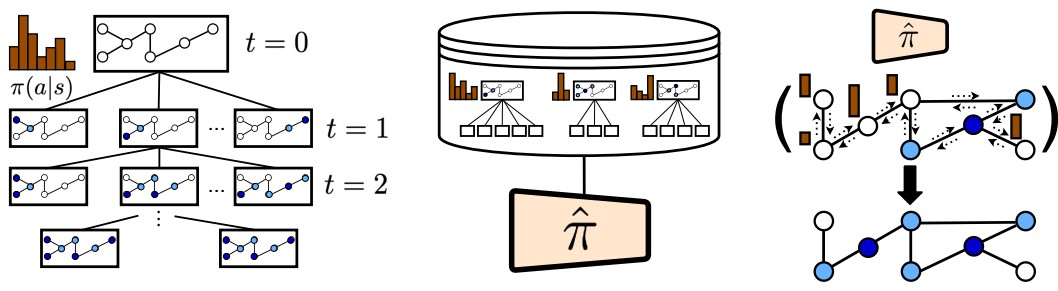

a) Collection of demonstrations by Monte Carlo Tree Search policy $\pi$

b) Training of GNN-parametrized policy $\hat{\pi}$ by imitation learning

c) Use of $\hat{\pi}$ to quickly predict on unseen game instances

Figure 1: Schematic of our approach for finding desirable equilibria in the graph-based best-shot game. (a) We exploit the correspondence between agents acquiring the public good in equilibria (pictured in dark blue above) and the Maximal Independent Set (mIS) structural property of graphs. We define an MDP that incrementally grows an independent set until it is maximal, and use Monte Carlo Tree Search (MCTS) to plan in this MDP in order to find desirable equilibrium configurations of the game. (b) We propose a Graph Imitation Learning method which uses demonstrations of the MCTS policy $\pi$ to learn a policy $\hat{\pi}$ parametrized by a graph neural network. (c) We use $\hat{\pi}$ to find optimal equilibrium configurations on unseen instances of the game.

corresponds to a Maximal Independent Set (mIS) [9]: a set of vertices of maximal size in which no two nodes are adjacent. Since enumerating mISs to identify desirable equilibria quickly becomes unfeasible for non-trivially sized graphs, practical alternatives are needed for larger graphs.

Towards this goal, Dall'Asta et al. [16] proposed a centralized algorithm based on best-response dynamics that converges to the optimal equilibrium (w.r.t. social welfare) in the limit of infinite time, and suggested a simulated annealing alternative. Levit et al. [38] proved that the general version of the best-shot PGG is a potential game and derived an algorithm for finding equilibria based on *side payments*, which are used by agents that are unhappy with their outcome to convince neighbors to switch. While superior results were obtained over best-response dynamics, there is still a wide gap between the equilibria found by this approach and optimal equilibria as found by exhaustive search on small graphs. Furthermore, current methods cannot optimize for criteria other than social welfare.

**Contributions.** Our contributions can be summarized as follows:

1. We propose to directly take advantage of the connection between equilibria in this class of games and Maximal Independent Sets. This relationship allows us to define an MDP which incrementally generates an mIS to optimize a desired property; thus, every configuration found is, by construction, an equilibrium of the game.[1] We adopt a variant of the Monte Carlo Tree Search algorithm for planning in this MDP. On small graphs, where an exhaustive enumeration of equilibria can be performed, the best outcomes found by this method are matched in most settings. On larger graphs existing methods are outperformed, especially in cases where the costs for acquiring the public good differ among players.

2. We devise a way to learn the structure of the solutions found by the planning algorithm based on imitation learning, such that predictions can be obtained on unseen game instances without repeating the search process. Specifically, we use a dataset of demonstrations of the search in order to learn a graph neural network parametrized policy through imitation learning, a procedure we call Graph Imitation Learning. The resulting policy is able to achieve 99.5% of the performance of the search method while being approximately three orders of magnitude quicker to evaluate and even exceeding the performance of the original search in some cases. This method is applicable beyond this class of networked public goods games, i.e., to a variety of graph-based decision-making problems where a model of the MDP is available and the goal is to maximize a graph-level objective function.

---

[1]We highlight the difference between *Maximal* Independent Set and *Maximum* Independent Set. A *Maximal* IS (mIS) is an IS that is not a proper subset of another IS. A *Maximum* IS (MIS) is an mIS of the largest possible size. In PGGs, an MIS may not be a desirable equilibrium, since it involves many players expending the cost.

## 2 Background and Related Work

**Network Games and Public Goods Games.** The literature on network (i.e., graph-based) games focuses on several fundamental questions regarding the behavior of agents that are connected by a network structure [28]: examples include proving the existence of and characterizing profiles of behavior in equilibria, reasoning in the presence of partial or probabilistic information [20], and examining the effect of new links in the graph [9]. Graphical games [30] represent a class of games on graphs in which actions are binary and utilities are defined in terms of neighbors in the graph structure. The networked PGG is a graphical game in which agent utilities are a function of aggregated neighbors' efforts, with the *best-shot* version being a sub-category in which utilities are a maximum of neighbor quantities. Recent works in this area treat the properties of binary PGGs [52] as well as designing strategies for inducing equilibria in such games by manipulating the graph structure itself [31].

**Computing Equilibria.** The problem of computing equilibria in non-cooperative games is well-studied. A large spectrum of techniques exist for 2-player and n-player games that find a sample equilibrium or enumerate all equilibria, with the task being computationally challenging or intractable in many cases [39, 17]. In graphical games, more efficient algorithms can be derived for restricted cases such as complete graphs or trees [30, 52]. For best-shot PGGs, best-response dynamics have been proven to converge to a PSNE [38], and as previously described several methods have been proposed for finding PSNEs [16, 38] that improve on standard best-response.

**Monte Carlo Tree Search.** For MDPs in which a model is available, lookahead can be used in order to select (*plan*) near-optimal actions. The Monte Carlo Tree Search (MCTS) family of algorithms uses returns estimated from sampled runs in order to make such decisions, with the UCT algorithm [35] formulating the decision at each node as a multi-armed bandit problem. UCT has been used to great success as a framework for challenging two-player zero-sum games such as Go [46]. Demonstrations of UCT have been used in order to train deep neural networks that mimic its policies while being much cheaper to evaluate [24]. Function approximators trained with this mechanism can be leveraged to bias the tree search, yielding an iterative procedure where both search and neural network improve [3, 47]. In the context of a different graph problem (namely, Knowledge Base Completion), M-Walk [45] also formulates the task as a deterministic MDP and uses MCTS in conjunction with a neural network parametrized policy.

**Learning and Search in Combinatorial Optimization.** In combinatorial optimization, the goal is to select an optimal solution among a large set of possible options. Graph-based combinatorial optimization problems such as the Traveling Salesperson Problem are well-studied, and due to their intractability [21] typically make use of approximate search or heuristics. Similarities among instances of such problems make them an attractive target for machine learning approaches [8], with this area of work gaining momentum in recent years [49, 6, 42]. Notably, Khalil et al. [33] proposed a framework for the discovery of combinatorial optimization heuristics using the DQN algorithm together with representations based on the *structure2vec* (S2V) graph neural network [14]. Various improvements on this scheme have been proposed, for example by allowing the agents to reverse decisions at runtime [5] or constructing several parts of the solution in parallel [2]. Alternatively, sample efficiency and generalization performance can be improved in some cases by combining a graph neural network with MCTS [1].

## 3 Notation and Problem Statement

**Game Definition.** A networked, best-shot public goods game takes place over an undirected, unweighted graph $G = (N, E)$ with no self-loops. Each vertex in $N = \{N_1, N_2, \ldots N_n\}$ represents a player, while edges $E$ capture the interactions between agents in the game. Each player chooses an action $a_i \in A_i$, where $A_i = \{0, 1\}$ denotes the action space of player $i$. We let action $1$ denote investment in the public good by the agent and $0$ denote non-investment. An action profile $\mathbf{a} = (a_1, \ldots, a_n)$ captures the choices of all players. The set $A = A_1 \times A_2 \cdots \times A_n$ denotes the set of all possible action profiles. We use $\mathbf{a}_{-i}$ to refer to actions of all other players except $i$, and $\mathbb{I}(\mathbf{a})$ to denote the set of all players that play action $1$ in action profile $\mathbf{a}$. Investment in the public good carries a cost $c_i \in (0, 1)$ for each player $i$. We use the terms *identical cost* (IC) to refer to the setting in which costs are the same for all players, and *heterogeneous cost* (HC) to refer to that in which costs are different between players. We let $\mathbf{c} = (c_1, \ldots, c_n)$.

We also define, for each player $i$, a *neighborhood* $\mathcal{N}_i$, which contains $i$ and all adjacent players, i.e., $\mathcal{N}_i = \{i\} \cup \{N_j \in N | (i, j) \in E\}$. The utility $u_i$ for player $i$ under an action profile $\mathbf{a}$ is defined as:

$$u_i(\mathbf{a}) = \begin{cases} 1 - c_i, & \text{if } a_i = 1 \\ 1, & \text{if } a_i = 0 \ \wedge \ \exists j \in \mathcal{N}_i \ . \ a_j = 1 \\ 0, & \text{if } a_i = 0 \ \wedge \ \forall j \in \mathcal{N}_i \ . \ a_j = 0 \end{cases}$$

We are interested in Pure Strategy Nash Equilibrium (PSNE) solutions, since there are no mixed strategy equilibria in this game [9]. A pure strategy is a complete, deterministic description of how a player will play the game. An action profile is a PSNE if all players would not gain higher utility by changing their action choice, given the actions of the other players. Formally, $\mathbf{a} \in A$ is a PSNE if and only if $u_i(a_i, \mathbf{a}_{-i}) \geq u_i(a_i', \mathbf{a}_{-i}) \ \forall i \in N, \ a_i' \in A_i$. The tuple $(G, A, u, \mathbf{c})$ defines an instance of this game. We let the set $\mathcal{E}$ denote all Pure Strategy Nash Equilibria of a game instance.

**Problem Statement.** We formulate our problem as follows: given a game instance $(G, A, u, \mathbf{c})$ and an objective function $f \colon \mathcal{E} \to [0, 1]$, the goal is to find the PSNE for which $f$ is maximized; concretely, finding $\mathbf{a}$ that satisfies $\mathrm{argmax}_{\mathbf{a} \in \mathcal{E}} f(\mathbf{a})$.

What constitutes a desirable equilibrium in this game? From a utilitarian perspective, a *desirable* equilibrium is one that *maximizes the social welfare* of agents. We define the $SW(\mathbf{a})$ objective as below, normalizing by the number of players $|N|$ so that games of different sizes are comparable:

$$SW(\mathbf{a}) = \frac{\sum_{i \in N} u_i(\mathbf{a})}{|N|}$$

A further desirable characteristic is *fairness*, or equality between players' utilities. We define the fairness objective $F(\mathbf{a})$ as the complement of the Gini coefficient, a measure of inequality:

$$F(\mathbf{a}) = 1 - \frac{\sum_{i \in N} \sum_{j \in N} |u_i(\mathbf{a}) - u_j(\mathbf{a})|}{2n \sum_{j \in N} u_j(\mathbf{a})}$$

**Maximal Independent Sets.** A subset of vertices $I \subseteq N$ is an Independent Set (IS) of the graph $G = (N, E)$ if none of the vertices in the set are adjacent to each other. Formally, $\forall i, j \in I$ s.t. $i \neq j \ . \ (i, j) \notin E$. A Maximal Independent Set (mIS) is an independent set that is not a proper subset of any other independent set. Bramoullé and Kranton [9] have proven a bidirectional correspondence between the set of players playing 1 in equilibria of the networked best-shot PGG and mISs. Thus, one way of finding desirable equilibria that is faster than considering all $2^n$ action profiles could be to enumerate all mISs. However, the best known family of algorithms [10] for this task has worst-case running time $O(3^{n/3})$, which makes even this approach impractical beyond very small graphs.

## 4 Proposed Method

The proposed approach, in contrast with previous methods, directly exploits the relationship between equilibria of this game and the Maximal Independent Set property. We assume that a central planner, with a global view of the graph, seeks to find optimal outcomes of this game. To this end, we formulate the construction of a mIS as a Markov Decision Process (MDP), in which an agent incrementally builds an IS, receiving a reward signal based on the objective $f$ once the IS is maximal. To plan in this MDP, we use the UCT algorithm. We also describe an imitation learning procedure based on behavioral cloning [44] that can be used to learn a generalizable model of equilibrium structure by mimicking the moves of the search. This policy can be evaluated rapidly on new instances of the game without performing a new search. Our method is illustrated in Figure 1 and detailed below.

### 4.1 MDP Definition

The MDP we propose is defined below and illustrated in Figure 2.

**State $\mathcal{S}$.** A state $S_t$ is a tuple $(G, I_t)$ formed of the graph and independent set $I_t$, with $I_0 = \varnothing$.

**Actions** $\mathcal{A}$. An action corresponds to the selection of a node that is not currently in the independent set. Given a state $(G, I_t)$, we define available actions as $\mathcal{A}_t = N \setminus \bigcup_{i \in I_t} \mathcal{N}_i$, i.e., nodes currently in the independent set and all their neighbors are excluded.

**Transitions** $\mathcal{P}$. Transitions are deterministic and correspond to the addition of a node to the independent set. Concretely, given that the agent selects node $a$ at time $t-1$, the next state is defined as $(G, I_t)$, where $I_t = I_{t-1} \cup \{a\}$.

**Rewards** $\mathcal{R}$. Rewards depend on the objective function $f$ considered (concretely, $SW$ or $F$), and are provided once an mIS is constructed, with all other intermediate rewards being 0.

**Terminal.** Episodes proceed until the agent has constructed an mIS and thus no valid actions remain.

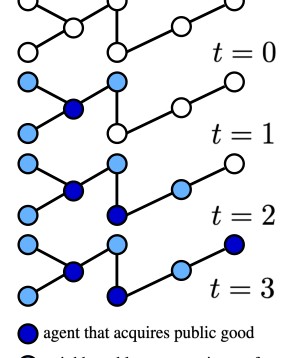

agent that acquires public good

neighbor able to access it cost-free

Figure 2: Illustration of the proposed MDP.

### 4.2 Collection of Demonstrations by Monte Carlo Tree Search

Since the MDP formulation above fully describes the transition and reward functions of this MDP, we may use model-based planning algorithms in order to plan an mIS that maximizes the desired objective. Concretely, we opt for the UCT [35] variant of the Monte Carlo Tree Search algorithm, which has proven to be an effective framework in a wide variety of decision-making problems. The UCT algorithm selects the child node corresponding to action $a$ that maximizes

$$UCT(s, a) = \frac{R(s, a)}{C(s, a)} + 2c_p \sqrt{\frac{2 \ln C(s)}{C(s, a)}},$$

where $R(s, a)$ is the sum of returns observed when taking action $a$ in state $s$, $C(s)$ is the visit count for the parent node, $C(s, a)$ is the number of child visits, and $c_p$ is a constant that controls the level of exploration [35]. We use the UCT policy $\pi$ to collect demonstrations using a set of training game instances $\mathbf{G}^{train}$, each of which contains $n$ players. This process is illustrated in Figure 1a. We let $\mathcal{D}_n$ denote this dataset of demonstrations. Each demonstration is effectively a datapoint and is composed of a tuple $(S_t, \mathcal{A}(S_t), n, \mathbf{v})$, where:

- $S_t$ represents the state from which the move is executed (i.e., for the mIS problem, the underlying graph $G$ and the current independent set $I_t$);
- $\mathcal{A}(S_t)$ represents the valid actions available at the state (i.e., it contains nodes that are not currently in $I_t$ or are a neighbor of any node in $I_t$);
- $n$ represents the size of the underlying graph (i.e., the number of nodes / players);
- $\mathbf{v}$ is a vector of size $|\mathcal{A}(S_t)|$, where each entry is equal to $C(S_t, a)$, for each valid action $a$. This represents the number of visits of the search policy from the root state $S_t$ to each of the possible actions $a$. Given that $C(S_t) = \sum_{a \in \mathcal{A}_t} C(S_t, a)$, the empirical policy is then estimated as $\pi(a|S_t) = \frac{C(S_t, a)}{C(S_t)}$.

### 4.3 Graph Imitation Learning

**Policy Parametrization.** The main disadvantage in using planning algorithms is that predictions are expensive to obtain for new game instances. To mitigate this, we explore the possibility of learning a policy $\hat{\pi}$ parametrized by a graph neural network (GNN). Specifically, we use the *structure2vec* GNN [14], which produces for each vertex $N_i \in N$ an embedding vector $\mu_{N_i}$ that captures the structure of the graph as well as interactions between neighbors. This is achieved in several rounds of combining the features of a node with an aggregation of its neighbors' features, to which an activation function is applied. Prior to this aggregation, node and neighbor features are multiplied by a set of parameters $\theta^{(1)}$. Embeddings for a state $S_t$ can be obtained by summing node embeddings: $\mu(S_t) = \sum_{N_i} \mu_{N_i}$. We use node features $\mathbf{x}_{N_i}$ corresponding to a one-hot encoding that captures whether the node is in the independent set, i.e., $\mathbf{x}_{N_i} = [1, 0]^T$ if $N_i \in I_t$ and $\mathbf{x}_{N_i} = [0, 1]^T$ otherwise.

An important challenge in this setting is the fact that at any timestep a significant number of actions are not available. Thus, the default choice of having the output layer consist of a softmax layer with

one unit per vertex in $N$ is potentially wasteful. In addition, such an architecture is sensitive to node relabeling and not transferable to other graphs. We thus consider a different approach: we make the final layer of the policy network output a *proto-action* $\phi(S_t) = \theta^{(2)}\text{relu}\,(\theta^{(3)}\mu(S_t))]$. Then, in order to obtain probabilities, we measure the Euclidean distances $d(a, \phi_t)$ between the proto-action and the embeddings of all available actions, normalized using a softmax with temperature $\tau$. This allows us to compute probabilities for all possible actions in a single forward pass. Formally:

$$\hat{\pi}(A_t|S_t) = \frac{\exp(d(\mu_{A_t}, \phi(S_t))/\tau)}{\sum_{a \in \mathcal{A}(S_t)} \exp(d(\mu_a, \phi(S_t))/\tau)} \tag{1}$$

**Loss Term.** For training the policy network, we minimize the KL divergence between the distribution of the policy network and the empirical distribution formed by the number of child visits [3], i.e.,

$$\mathcal{L} = -\sum_{a \in \mathcal{A}(s)} \frac{C(s,a)}{C(s)} \log(\hat{\pi}(a|s)) \tag{2}$$

**Training Strategies.** We consider several training strategies for obtaining a model for a target size $n$ of game instances (i.e., games in which there are $n$ players). Since the structure of equilibria for increasingly large number of players are potentially more complex (i.e., we have to deal with larger graphs), we also consider whether training additionally on smaller graphs (thus simpler examples) brings a generalization benefit. We employ *curriculum learning*, a methodology successful in a variety of machine learning settings [7, 53], and compare against training only on the target size as well as mixing the examples of different sizes instead of constructing a curriculum. Concretely, we consider the following training strategies, noting that validation is performed on the target size $n$:

- **separate**: train only on examples from $\mathcal{D}_n$;
- **mixed**: train on examples from $\bigcup_{m \leq n} \mathcal{D}_m$;
- **curriculum**: carry out the training in several epochs, at each epoch considering only examples from $\mathcal{D}_m$, with each value $m \leq n$ considered in ascending order.

We learn the parameters $\Theta = \{\theta^{(i)}\}_{i=1}^3$ as well as the softmax temperature $\tau$ in an end-to-end fashion. When evaluating this policy, we use greedy action selection. Our method, which we refer to as *GIL* (Graph Imitation Learning), is illustrated in Figure 1b. A pseudocode description is given in Algorithm 1 in the Appendix.

## 5 Experiments

### 5.1 Evaluation Procedure

**Game Instances.** To evaluate our approach and all baselines, we create instances of network games over graphs with a number of players $n \in \{15, 25, 50, 75, 100\}$. For each size and each underlying graph model, we generate: $10^3$ training instances $\mathbf{G}^{train}$; $10^2$ validation instances $\mathbf{G}^{eval}$ used for hyperparameter optimization; and $10^2$ test instances $\mathbf{G}^{test}$. To set costs $\mathbf{c}$, in the IC case we fix $c_i = 1/2$, $\forall i$, while for HC we consider costs uniformly sampled in $(0, 1)$. To generate the underlying graphs $G$ over which the game is played, we use the following synthetic models:

- *Erdős–Rényi (ER)*: A graph sampled uniformly out of all graphs with $n$ nodes and $m$ edges [18]. We use $m = \frac{20}{100} * \frac{N*(N-1)}{2}$, which represents 20% of all possible edges.
- *Barabási–Albert (BA)*: A growth model where $n$ nodes each attach preferentially to $M$ existing nodes [4]. We use $M = 2$.
- *Watts–Strogatz (WS)*: A model designed to capture the small-world property found in many social and biological networks, which generates networks with high clustering coefficient [51]. Starting with a regular ring lattice with $n$ vertices with $k$ edges each, edges are rewired to a random node with probability $p$. We use $k = 2$ and $p = 0.1$.

**Baselines.** We compare to the following baselines that have been proposed in prior work:

Table 1: Mean rewards obtained by the methods split by cost setting, graph model, and objective.

| c | G | f | Random | TH | TLC | BR | PT | SA | UCT | GIL (ours) |
|---|---|---|---|---|---|---|---|---|---|---|
| HC | BA | F | $0.745_{\pm0.005}$ | 0.802 | 0.774 | $0.742_{\pm0.004}$ | $0.791_{\pm0.015}$ | $0.815_{\pm0.000}$ | $\mathbf{0.837}_{\pm0.000}$ | $0.834_{\pm0.001}$ |
| | | SW | $0.697_{\pm0.007}$ | 0.779 | 0.727 | $0.691_{\pm0.006}$ | $0.760_{\pm0.019}$ | $0.795_{\pm0.000}$ | $\mathbf{0.815}_{\pm0.000}$ | $0.813_{\pm0.000}$ |
| | ER | F | $0.877_{\pm0.001}$ | 0.896 | 0.920 | $0.877_{\pm0.000}$ | $0.911_{\pm0.002}$ | $0.908_{\pm0.001}$ | $\mathbf{0.945}_{\pm0.000}$ | $0.940_{\pm0.003}$ |
| | | SW | $0.868_{\pm0.001}$ | 0.890 | 0.912 | $0.867_{\pm0.000}$ | $0.903_{\pm0.002}$ | $0.903_{\pm0.001}$ | $\mathbf{0.940}_{\pm0.000}$ | $0.935_{\pm0.001}$ |
| | WS | F | $0.803_{\pm0.002}$ | 0.806 | 0.865 | $0.804_{\pm0.002}$ | $0.821_{\pm0.003}$ | $0.832_{\pm0.001}$ | $\mathbf{0.892}_{\pm0.000}$ | $\mathbf{0.892}_{\pm0.000}$ |
| | | SW | $0.781_{\pm0.002}$ | 0.785 | 0.846 | $0.782_{\pm0.003}$ | $0.800_{\pm0.004}$ | $0.817_{\pm0.001}$ | $\mathbf{0.876}_{\pm0.000}$ | $\mathbf{0.876}_{\pm0.000}$ |
| IC | BA | F | $0.833_{\pm0.000}$ | 0.844 | — | $0.834_{\pm0.000}$ | $0.841_{\pm0.005}$ | $\mathbf{0.849}_{\pm0.000}$ | $0.847_{\pm0.000}$ | $0.847_{\pm0.000}$ |
| | | SW | $0.697_{\pm0.007}$ | 0.779 | — | $0.691_{\pm0.006}$ | $0.757_{\pm0.019}$ | $0.794_{\pm0.000}$ | $\mathbf{0.795}_{\pm0.000}$ | $\mathbf{0.795}_{\pm0.000}$ |
| | ER | F | $0.893_{\pm0.000}$ | 0.906 | — | $0.892_{\pm0.000}$ | $0.907_{\pm0.001}$ | $0.916_{\pm0.000}$ | $\mathbf{0.922}_{\pm0.000}$ | $0.919_{\pm0.002}$ |
| | | SW | $0.867_{\pm0.000}$ | 0.889 | — | $0.866_{\pm0.001}$ | $0.889_{\pm0.002}$ | $0.903_{\pm0.000}$ | $\mathbf{0.910}_{\pm0.000}$ | $0.908_{\pm0.001}$ |
| | WS | F | $0.842_{\pm0.001}$ | 0.843 | — | $0.842_{\pm0.001}$ | $0.847_{\pm0.001}$ | $0.856_{\pm0.000}$ | $0.862_{\pm0.000}$ | $\mathbf{0.864}_{\pm0.000}$ |
| | | SW | $0.777_{\pm0.002}$ | 0.782 | — | $0.779_{\pm0.003}$ | $0.791_{\pm0.004}$ | $0.813_{\pm0.001}$ | $0.824_{\pm0.000}$ | $\mathbf{0.828}_{\pm0.000}$ |

- *Exhaustive Search (ES)*: Select the PSNE that maximizes the objective out of $2^n$ possible action profiles. Only applicable on very small graphs due to its computational complexity.

- *Best Response (BR)*: In the graph-based best-shot PGG, Best Response converges to a PSNE [38]. We start from a randomly selected action profile, allow the agents to iteratively play Best Response, and measure $f$ once a PSNE is reached.

- *Payoff Transfer (PT)*: The method of Levit et al. [38] modifies the definition of utilities in this game to include an additional term that represents a payoff. The distributed procedure they propose enables agents to convince their neighbors to switch their action by providing a payoff. As with BR, we start from a randomly selected action profile, and measure $f$ once a PSNE is reached. Even though the PSNEs reached do not necessarily correspond to mISs, we evaluate the objective functions by considering the utilities of the players.

- *Simulated Annealing (SA)*: The method proposed by Dall'Asta et al. [16] works by randomly selecting an agent playing $0$, incentivizing them to switch their action to $1$, then iterating on the BR rule. The new PSNE reached is either accepted or rejected based on a simulated annealing rule with a certain temperature parameter $\epsilon$. In the limit of infinite time, this method converges to the optimal Nash equilibrium in terms of social welfare in the IC case.

We also consider the additional baselines listed below, which exploit the mIS connection. We note that TH and TLC have not been considered in prior work but are potentially effective heuristics.

- *Random*: Incrementally construct an mIS by randomly picking, at each step, a node that is not in the IS and is not adjacent to any nodes in the IS.

- *Target Hubs (TH)*: Pick the highest-degree node available that is not yet included in the IS. While this strategy is not guaranteed to find a global maximum, placing the public good on nodes with many connections means that many others can access it.

- *Target Lowest Cost (TLC)*: Place the public good on nodes for which the cost $c_i$ is lowest (only applicable in HC case). This may result in equilibria with high social welfare and fairness since the good is acquired only by those players for which it is cheap to do so.

**Training and Evaluation Protocol.** Evaluation (and training, where applicable) is performed separately for each $n$, graph model, objective $f$, and cost setting (IC or HC). To compute means and confidence intervals, we repeat the evaluation (and training) across 10 different random seeds.

**Hyperparameters.** We optimize hyperparameters for UCT, GIL, and the SA methods; the other methods are hyperparameter-free. For UCT, we use a random simulation policy and a number of node expansions per move $n_{sims} = 20n$ (larger values provide diminishing returns). At each step, once simulations are completed, we select the child node with the largest number of visits as the action (ROBUSTCHILD). We treat $c_p$ as a hyperparameter to be optimized, and for each problem instance we consider $c_p \in \{0.05, 0.1, 0.25, 0.5, 1, 2.5\}$. Since the ranges of the rewards may vary in different settings, we further standardize $c_p$ by multiplying with the average reward $R(s)$ observed at the root in the previous timestep – ensuring consistent levels of exploration. For GIL, the learning rate $\eta \in \{10^{-2}, 10^{-3}, 10^{-4}\}$, the number of S2V message passing rounds $K \in \{3, 4, 5, 6\}$, and the

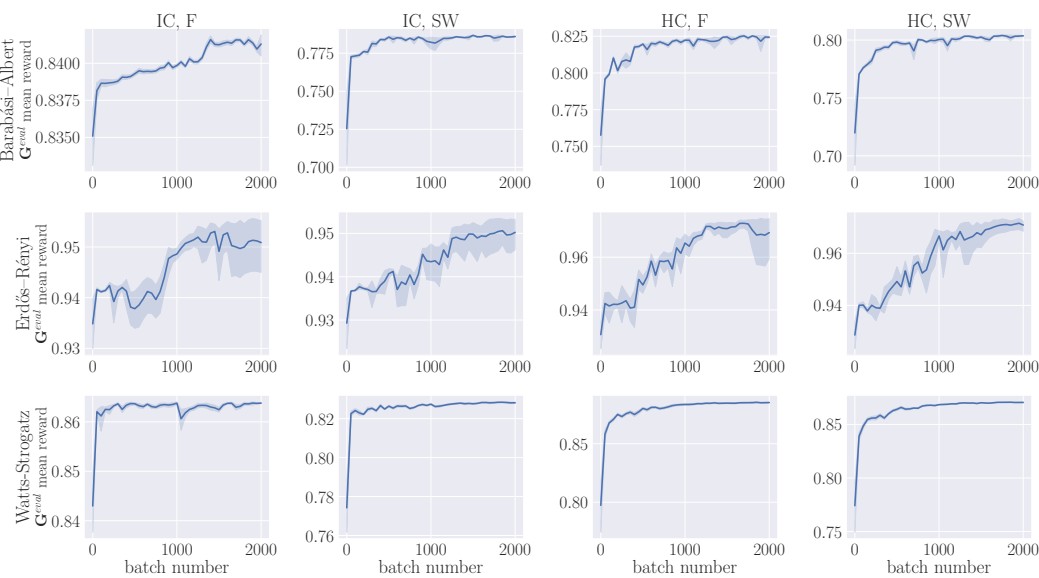

Figure 3: Training curves for GIL, showing performance on the held-out validation set.

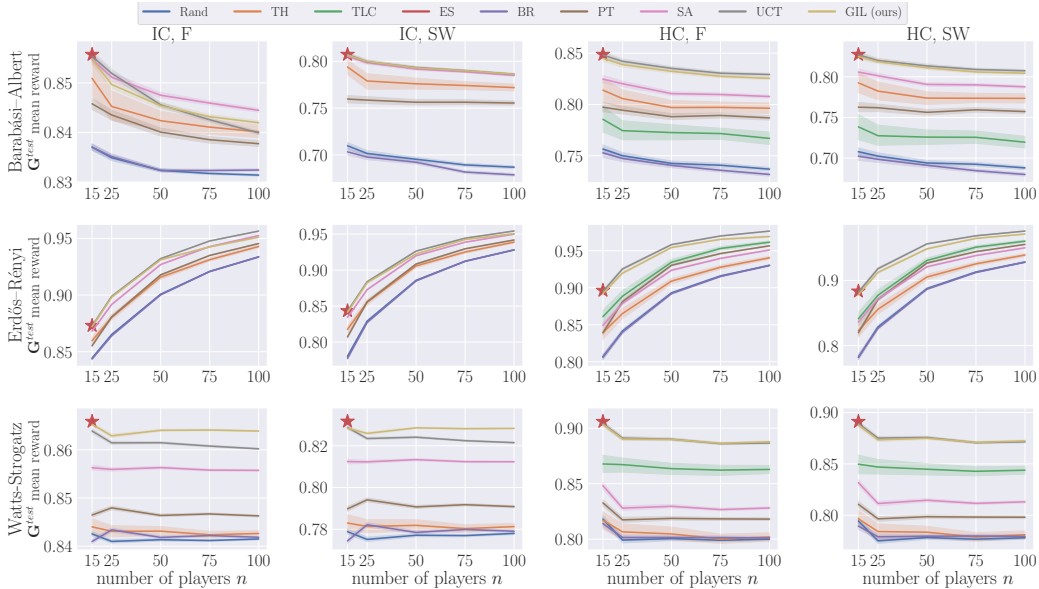

Figure 4: Mean rewards obtained by the methods as a function of the number of players $n$.

training strategy (separate, mixed, or curriculum) are optimized using a grid search. For SA, we consider $\epsilon \in \{10^1, 10^2, 10^3, 10^4\}$, stop the optimization after $10^4$ steps without an improvement, and use a cut-off of $10^7$ steps. Further details are provided in the Appendix.

**GIL Training.** GIL is the only method that requires training. The datasets on which this method is trained correspond to demonstrations of the hyperparameter-optimized UCT. We carry out the imitation learning procedure as described in Section 4.3 and evaluate performance on the validation instances every 50 steps. We train using the Adam [34] optimizer for $2 \times 10^3$ steps and use a batch size of 5 in all cases (larger batch sizes proved harmful). The dimension of the proto-action vector $\phi$ and the number of S2V latent variables are both 64. The temperature $\tau$ is initialized to 10.

**Implementation.** Our implementation is available at `https://github.com/VictorDarvariu/solving-graph-pgg`. Please consult the Appendix for more details.

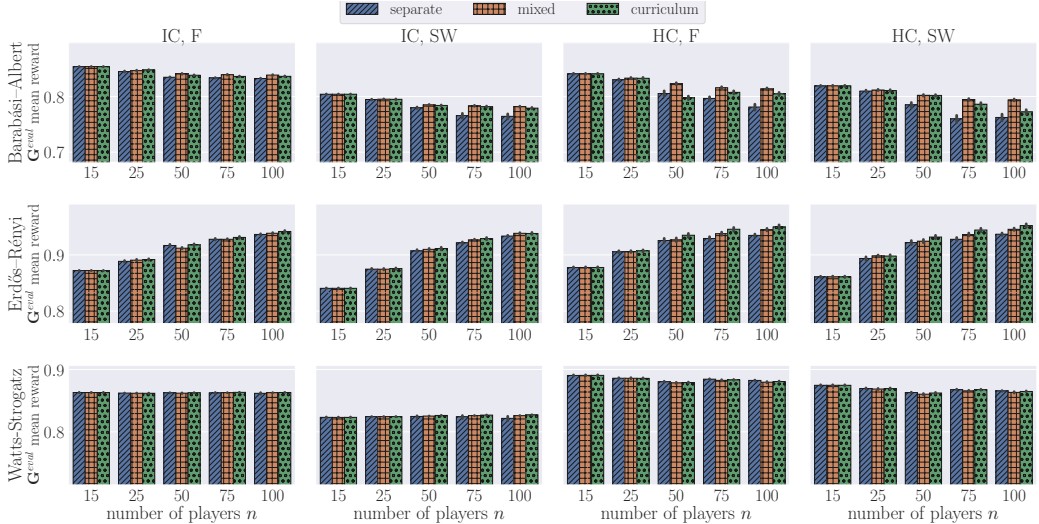

Figure 5: Mean rewards obtained on the validation set by GIL using different training procedures. The choice of an appropriate procedure depends on the underlying graph structure. We observe that it is more important in the HC case and it increases in importance with graph size.

## 5.2 Evaluation Results

We show the main results obtained in Table 1, in which entries are aggregated across game sizes $n \in \{15, 25, 50, 75, 100\}$ (for a version in which they are separated, consult Table 2 in the Appendix). Each reported value corresponds to the average objective function value of an equilibrium of the graph-based best-shot public goods game. Performance obtained during training on the held-out validation set of instances with $n = 100$ is shown in Figure 3. The performance on the test set is shown in Figure 4, in which the $x$-axes represent the number of players. We also show a comparison of the runtime per episode in milliseconds used by the different methods in Figure 6. In the figures, the stars at $n = 15$ represent exhaustive search.

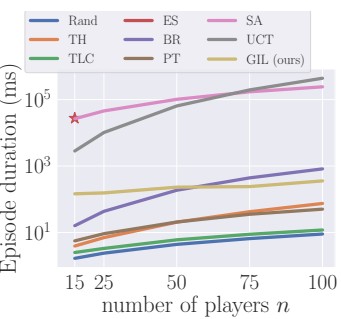

Figure 6: Mean milliseconds needed to complete an episode (i.e., construct an mIS) as a function of the number of players.

We find that the UCT planning method outperforms previous methods in all cases except the $IC, F$ case in which the SA baseline does better as game size increases. For the smallest graphs, UCT nearly performs on par with exhaustive search. The Random and BR baselines consistently perform the poorest, as expected. The gap between the methods enabled by our approach (UCT and GIL) are higher in the HC settings where costs for acquiring the public good differ between players.

**Imitation Learned Policy.** We find that the performance of the imitation learned policy GIL is very close to that of the planning method (at least 99.5%), even clearly exceeding it in certain cases (in the IC setting for Watts-Strogatz graphs and both objectives). Given the timings in Figure 6, this closeness in performance is even more remarkable since the imitation learned policy is approximately three orders of magnitude cheaper to evaluate than the planning method on the largest graphs tested.

**Impact of Training Strategies.** Additionally, we also explore the impact of the training strategies used for the imitation learning phase in Figure 5. Since the choice of which examples to present to the model as well as their order could have an impact on the performance of the imitation-learning trained policy and there is no a priori knowledge of which strategy is optimal, we treat the training strategy as a tunable hyperparameter. We find that no training strategy is better across the board; rather, their rankings are consistent depending on the graph model on which the method is trained. For BA graphs, the mixed strategy performs best; in the case of ER graphs the curriculum strategy achieves the highest reward; for WS the differences between the averages are very small. Additionally,

for BA and ER graphs, the relative differences between the methods are higher in the HC setting than in IC. In all cases, the difference in mean reward for the different training strategies is insignificant on small graphs, and increases in importance the larger the size.

## 6 Discussion

**Limitations.** Since the proposed method exploits the connection with the mIS property, it is only applicable for the class of games for which it holds, and cannot be directly applied to a wider class of networked public goods games. In addition, given that this method assumes a centralized perspective, it is only applicable in situations in which the network structure is known or can be reasonably inferred. Even though the work in this paper answers the question of what outcomes *could* be reached by self-interested players, the design of mechanisms to move towards such configurations remains a challenging problem. Possible incentivization mechanisms that we aim to explore in future work include incentivizing individual players as well as modifying the network structure itself. Since the impact of such interventions on players' actions can be captured by best-response dynamics, this can be formulated as a search problem in which the goal is to find the set of interventions that brings us arbitrarily close to the goal state. At a high level, an approach similar to the proposed method, which leverages a different deterministic MDP model capturing the impact of interventions, may be used.

**Societal Impact and Implications.** The direct implication of this work is that it enables a social planner to find beneficial outcomes that can be achieved and maintained by self-interested players for situations that can be modeled by networked best-shot games. This class of games is relevant for a variety of scenarios in which the agents forming a society can choose to contribute effort to a public good, and our work is motivated by positive societal impact. We cannot foresee situations in which this method can be directly misused. This relies, however, on the assumption that the social planner aims to improve outcomes, rather than have a negative impact. Furthermore, this type of modeling is necessarily abstract and makes simplifying assumptions that may not hold in the real world.

## 7 Conclusion

In this work, we have considered the problem of finding desirable equilibria of the graph-based best-shot public goods game. We have approached this from the perspective of a principal agent with global knowledge of the game, who aims to find optimal Pure Strategy Nash Equilibria in terms of social welfare and fairness of outcomes. We have defined an MDP to find such equilibria, which relies on the connection with the Maximal Independent Set structural property of graphs. Using the UCT algorithm to plan in this MDP yields better results than existing approaches, especially in the case where the costs of acquiring the public good differ between players. We have also proposed a Graph Imitation Learning method which is able to learn the structure of these equilibria, yielding performance within 99.5% of the planning method while generalizing to different game instances and generating predictions approximately three orders of magnitude quicker on the largest graphs tested.

The proposed method is directly applicable to other settings in which mIS are of interest (see e.g. [15]). More broadly, the method for performing planning and imitation learning presented in this work is applicable to a variety of problems which can be formulated as a decision-making process on a graph with the goal of maximizing a given objective function. Areas where this may be of interest include combinatorial optimization and algorithmic reasoning over graphs [8, 11], provided that the horizon for the task is manageable by a search procedure. While we have focused on constructing generalizable models, if predictions need to be made for a single graph instance, one could also consider combining the planning and imitation learning step in an iterative method similarly to the ExIt algorithm [3, 47]. This work is related to other recent work that considers learning in network games (e.g., [48] treats network *emergence* games) as well as more broadly to ongoing initiatives to study cooperation in multi-agent systems and its impact on societal problems [13].

## Acknowledgments and Disclosure of Funding

This work was supported by The Alan Turing Institute under the UK EPSRC grant EP/N510129/1. The authors declare no competing financial interests with respect to this work.

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
