# OpenReview forum: "Solving Graph-based Public Goods Games with Tree Search and Imitation Learning"
_NeurIPS.cc/2021/Conference — NeurIPS 2021 Poster_

### Official Review · Reviewer_cDVM · 2021-07-12

**Rating:** 6
**Confidence:** 3

**Summary:**

The problem studied by the authors is finding pure Nash equilibria in public goods games on graphs that maximize certain functions (e.g., social welfare, fairness). The authors search over maximal independent sets on the graph, a connection that was proven equivalent to PSNE by previous authors. Past approaches include exhaustive search, iterative best response, and several bespoke algorithms developed over the past 20 years. The authors formulate the search problem as an MDP, where the maximal independent set is formed by adding one node at a time. The authors solve this search problem in two ways. The first is with out-of-the-box Monte Carlo tree search (UCT). The second is via imitation learning, where a GNN is trained to predict the best decisions in Monte Carlo tree search. The first, UCT, approach achieves strong performance w.r.t. mean reward across many instances, beating the benchmarks. The imitation learning approach also beats the benchmarks and run several orders of magnitude faster once trained. Both methods appear to lie on the Pareto frontier of speed vs. performance, in the slower speed/higher performance part of the space.

**Limitations And Societal Impact:**

The authors have a para devoted to limitations and one devoted to social impact. Both are clear and useful. I would suggest moving the social impact para to the supplement if more space is needed—the work is quite abstract, and I did not find social impact to be a major concern.

**Main Review:**

The authors' approach falls under the broad umbrella of applying AI/ML techniques to combinatorial optimization. This is not a new idea in itself, but there are not that many examples where it has looked as useful as it does here. The paper is well written and clear overall. I have a few specific comments:
- The authors should include a formal description of their algorithms in pseudocode, either in the main text or supplement. This is particularly an issue for the imitation learning approach, which is complex and not that clearly described in the text.
- The authors should report more statistics than average reward and standard error in experiments. In particular, they should report some kind of win/loss metric, potentially in the supplemental material. The issue is that some methods could be outperforming others by a large amount on a small number of problems, but by a lot, and still be performing worse on many.
- I could not find a clear report of the time spent training IL-UCT to get the level of performance they achieve. Training curves for IL-UCT and time per iteration need to be included.

I thought the method of selecting actions via the proto-action was interesting and the authors could advertise this contribution more (say in a sentence in the intro).

Post-response: I appreciate the authors' response. It seems there are no major points of disagreement.

**Time Spent Reviewing:**

4

---

> ### Author Response · Authors · 2021-08-09
> **Response to Reviewer cDVM**
>
> We would like to thank the reviewer for carefully considering our work and for providing us with their detailed comments. We appreciate the very useful feedback, which will help to improve the paper significantly, especially in terms of presentation clarity and robustness of the results. We include below our responses to the major comments.
>
> > The authors should include a formal description of their algorithms in pseudocode, either in the main text or supplement. This is particularly an issue for the imitation learning approach, which is complex and not that clearly described in the text.
>
> We agree that  providing the readers with a self-contained pseudocode description would be helpful given the complexity of the method. As the reviewer probably noticed, the various components are explained throughout the manuscript. The only component that has not been covered is the dataset collection process, which we have now explained in our response to reviewer Ti5Q. But we agree that this is confusing. We will revise the paper in order to present the full algorithm if accepted.
>
> > The authors should report more statistics than average reward and standard error in experiments. In particular, they should report some kind of win/loss metric, potentially in the supplemental material. The issue is that some methods could be outperforming others by a large amount on a small number of problems, but by a lot, and still be performing worse on many.
>
> We agree that including an additional metric that discounts this possibility is worthwhile, and would improve confidence in the results. Since a win/loss metric is not directly applicable in this setting, we think that a ranking based metric, which compares the relative performance of the different methods for many problem instances, should capture this comparison well without being influenced by the scale of the differences. Given the gaps observed relative to the other methods and the relatively small scales of the differences (e.g., Figure 3), we expect the results to hold, but this is a check that should be performed.
>
> > I could not find a clear report of the time spent training IL-UCT to get the level of performance they achieve. Training curves for IL-UCT and time per iteration need to be included.
>
> We have logged the training curves for IL-UCT during our experimental runs, and are indeed happy to include the discussion of these results. In summary, it is possible to observe that, even on the largest graphs, the validation performance increases very sharply after about 50 batches each of size 5, and continues to slowly increase or fluctuate after that. The model is thus very cheap to train, even without a GPU -- as reported in the supplementary material, we have trained the models exclusively on CPUs.
>
> > I thought the method of selecting actions via the proto-action was interesting and the authors could advertise this contribution more (say in a sentence in the intro).
>
> Indeed, we also empirically found this architecture to be very data-efficient. We have not explicitly brought it forward since it is somewhat orthogonal to the application to public goods games, but we agree that it should be highlighted more in the paper and we will modify the text accordingly if this work will be accepted. For additional context, this is also discussed in our reply to Reviewer ExWG in depth.

---

### Official Review · Reviewer_Ti5Q · 2021-07-16

**Rating:** 6
**Confidence:** 2

**Summary:**

This paper tries to solve the public good game problem (NP-complete). Existing methods can only find sub-optimal results of this problem and cannot optimize in different criteria. This paper proposes a method that directly uses the relationship between equilibria of the public good game problem and the Maximal Independence Set problem. The proposed method outperforms existing approaches in searching for the best equilibria. The paper uses imitation learning technique to generalize the learned equilibria to unseen game instance effectively.

**Limitations And Societal Impact:**

The authors have adequately addressed the limitations and potential negative societal impact of their work.

**Main Review:**

Originality: First, this paper proposes a new method to solve the classic public good game problem. It uses the correspondence of this problem and maximal independence set problem to define a Markov decision process. And by doing this, it adopts a variant of Monte Carlo Tree Search method to solve the planning problem in this MDP. The searched result is the wanted equilibria in the public good game problem. In a word, this paper transforms the public good game problem into a search problem and uses Monte Carlo Tree search method to solve it. Second, this paper use imitation learning to learn the structure of the equilibria found by the proposed MDP, such that it could predict on novel game instances without repeating the search process. It uses a dataset of demonstration of the Monte Carlo Tree Search to train a graph neural network and this GNN represented a policy learn from the dataset.

Quality: the paper shows that the proposed method outperforms many baseline methods. It has higher mean rewards on many game instances and the outperformance holds as the size of the problem increases. The proposed method did run faster than some baseline methods.

- As for the imitation learning part (section 4.3), the description of Figure 4 is not very clear. What is the use of that figure to explain the quality of the imitation learning part?

- As for the collection of demonstration part (section 4.2), how does the demonstration look like and how to collect it to be the dataset is not stated very clearly. Providing more details might help the reviewer to have a more straightforward understanding in the following imitation learning part.

Clarity: this paper is well organized. It first introduces public good game problem clearly and explain many infrequent notion and terms used in the paper, so it won’t cause too much difficulty in reading this paper. The proposed method part is well organized and easy to follow.

Significance: This paper is proposing an important method. First, it can be used to solve the public good game problem that outperforms existing methods. Second, the problem it tries to solve is very hard (NP-complete) and the current method (including the proposed one) can only reach a sub-optimal solutions.



**Time Spent Reviewing:**

3 hours

---

> ### Author Response · Authors · 2021-08-09
> **Response to Reviewer Ti5Q**
>
> We would like to thank the reviewer for carefully considering our work and for providing us with their detailed comments. We appreciate their feedback and structured discussion of the originality, quality, clarity and significance of the manuscript. We include below our responses to the major comments made by the reviewer.
>
> > As for the imitation learning part (section 4.3), the description of Figure 4 is not very clear. What is the use of that figure to explain the quality of the imitation learning part?
>
> As described in Section 4.3, there are several possible choices in terms of how the dataset of search demonstrations is used to train the imitation learning model. Our hypothesis is that, as the size of the game instances increases, both the choice of which examples to present to the model as well as the order in which they are presented could have an impact of the performance of the imitation-learning trained policy. We have thus considered (Section 4.3, lines 202-216) the _separate_, _mixed_, and _curriculum_ training strategies. Since we have no a-priori knowledge of which strategy is optimal, and as their impact may depend on cost setting as well as the characteristics of the graph (size and structural properties), we have treated the strategy as a hyperparameter that can be tuned. __Figure 4 represents an exploration of the impact of the training strategy hyperparameter on the performance of the imitation-learned policy__, which indeed validates our hypothesis that different training strategies yield better or worse performances depending on cost setting, network size and characteristics. Choosing this hyperparameter appropriately is key to being  able to match the search performance on the larger graphs -- for further evidence of this, Table 2 in the supplementary material can be consulted. Our conclusions regarding the impact of this hyperparameter are presented in Section 5.2, lines 305-312, which discusses the results in Figure 4. We include them here for completeness:
>
> *"We find that no training strategy is better across the board; rather, their rankings are consistent depending on the graph model on which the method is trained. For BA graphs, the mixed strategy performs best; in the case of ER graphs the curriculum strategy achieves the highest reward; for WS the differences between the averages are very small. Additionally, for BA and ER graphs, the relative differences between the methods are higher in the HC setting than IC. In all cases, the difference in mean reward for the different training strategies is insignificant on small graphs, and increases in importance the larger the size."*
>
> We will revise the paper to discuss this aspect in more detail and better link the figure to the sections to which it is relevant.
>
> > As for the collection of demonstration part (section 4.2), how does the demonstration look like and how to collect it to be the dataset is not stated very clearly. Providing more details might help the reviewer to have a more straightforward understanding in the following imitation learning part.
>
> Indeed, there are details that we have left out due to space limitations, but we agree that they are important in order to understand the method more clearly. Following the notation given in the paper, each search _demonstration_ is composed of a tuple $(S_t, \mathcal{A}_t, n, \mathbf{v})$, where:
> - $S_t$ represents the state from which the move is executed (i.e., the underlying graph $G$ and the current independent set $I_t$);
> - $\mathcal{A}_t$ represents the valid actions available at the state (i.e., it contains nodes that are not currently in $I_t$ or are a neighbor of any node in $I_t$);
> - $n$ represents the size of the underlying graph (i.e., the number of nodes / players);
> - $\mathbf{v}$ is a vector of size $|\mathcal{A}_t|$, where each entry is equal to $C(S_t, a)$, for each valid action $a$. This represents the number of visits of the Monte Carlo Tree Search from the root state $S_t$ to each of the possible actions $a$.
>
> From the demonstration, given that $C(S_t) = \sum_{a \in \mathcal{A_t}} {C(S_t, a)}$, the (empirical) policy is then $\pi(a|S_t) = \frac{C(S_t,a)}{C(S_t)}$. This information represents one "data point" that is used for training.
>
> We describe the dataset collection below in pseudocode, noting that, as mentioned in Section 5.1, for all experiments we use 1000 training graphs:
>
> ```
> dataset = list()
> for each training graph:
> 	independent_set = set()
> 	valid_actions = nodes(graph)
> 	while length(valid_actions) > 0:
> 		demonstration = run_MCTS(graph, independent_set)
> 		dataset.add(demonstration)
>
> 		action = max_count_action(demonstration)
>
> 		independent_set.add(action)
> 		valid_actions.remove(action)
> 		for neighbor in neighbors(graph, action):
> 			valid_actions.remove(neighbor)
>
> ```
>
> We hope these explanations might be helpful for understanding the details of the data collection process. We will revise the paper to include the discussion above.

---

### Official Review · Reviewer_ExWG · 2021-07-16

**Rating:** 6
**Confidence:** 4

**Summary:**

This paper algorithms for computing desirable equilibrium strategies (maximize social welfare or fairness) in best-shot public good games (PGGs), which are PGGs on graphs in which each agent wants itself (good) or one of its neighbors (best) to have a good (though obtaining the good itself comes with a cost).  Initially, the paper describes an algorithm for computing equilibria of such games by defining and MDP constructed by identifying Maximal Independent Sets.  The resulting algorithms performs well, particularly on large graphs.  The paper also proposed a way to learn the structure of the solutions found by the planning algorithm based on imitation learning, which achieves 99.5% performance will reducing the computation time by approximately 3-orders of magnitude.

**Limitations And Societal Impact:**

Yes.

**Main Review:**

Strengths:

+ The problem definition, solutions, and results are describe carefully, rigorously, and clearly, such that is it easy to follow the paper.

+ The outcomes of the algorithms are good, exceeding the state-of-the-art, and the analysis is extensive.

+ The paper is presented in a way that it is enjoyable to read.

Questions about significance:

- PGGs are interesting because the group good is somewhat in conflict with individual incentives.  Computing desirable solutions in these games from a centralized perspective may be useful, as it defines what society could do.  However, as the paper acknowledges, it is unclear how to get individuals to actually carry out the computed solution.  So I in some sense question the papers usefulness (perhaps because I lack some domain knowledge or an imagination).

- As acknowledged in the paper, the paper’s scope is limited in that it only works for this very specific set of games.

In short, the paper is well-done in all respects.  My questions regarding the work lie in its significance to the NeurIPS community, but I hesitate somewhat to maintain that position (as it seems somewhat unfair).

Irrelevant nitpick:
- Figure 4 is discussed in the test after Figure 5 (I think).  Perhaps reorder the figures?

**Time Spent Reviewing:**

?

---

> ### Author Response · Authors · 2021-08-09
> **Response to Reviewer ExWG**
>
> We would like to thank the reviewer for carefully considering our work and for providing us with their detailed comments. We are glad that they found the presentation of our paper to be rigorous and clear, and they underlined the fact that the algorithms show good performance. We include below our responses to the major comments made by the reviewer.
>
> > PGGs are interesting because the group good is somewhat in conflict with individual incentives. Computing desirable solutions in these games from a centralized perspective may be useful, as it defines what society could do. However, as the paper acknowledges, it is unclear how to get individuals to actually carry out the computed solution. So I in some sense question the papers usefulness (perhaps because I lack some domain knowledge or an imagination)
>
> As discussed in the manuscript, we agree that getting the individuals to carry out the solution is not straightforward. We did not discuss this in depth due to space limitations, but in hindsight it is an important aspect to be considered in order to show the potential applicability of the proposed method. We hope that the discussion below will offer some clarifications.
>
> We think that *computing* the goal state that we would like to reach is necessarily the first step towards improving the society-level outcomes. The alternative to this is to carry out many interventions and "learn by experience" how these impact the society, in the hope that eventually a better outcome will be reached. Such interventions are in principle expensive to carry out, and as such, we consider that knowing the goal state a priori can significantly improve outcomes if we have "few shots" at intervening in the society.
>
> Thus, getting the individuals to carry out the solution essentially becomes a (different) search problem: finding the set of interventions that would bring us from the current configuration of the society to the goal state (or arbitrarily close). In theory, this can also be carried out in simulation since, for this class of games, the rational behavior of the agents is captured by best-response dynamics -- and thus we have a mechanism for modeling what the impact of _one_ intervention would be. Certainly, a _set_ of interventions may interact in non-trivial ways, and navigating the search space becomes complex. We think that, at a high level, an approach similar to the one proposed in the paper is also applicable for this reverse search problem. Specifically, using the fully-specified MDP model to carry out a tree search should provide significant gains over a brute-force or heuristic search. The tree search itself can be generalized by a graph neural network with our proto-action architecture.
>
> We hope that the above offers a relatively clear path towards the solution of the problem raised by the reviewer: we will modify the manuscript to discuss this in depth.
>
>
> > As acknowledged in the paper, the paper’s scope is limited in that it only works for this very specific set of games
>
> Indeed, within the realm of network games, our method is only applicable to a specific class of games. However, as discussed in the conclusion section (lines 341-346), the method is directly applicable, without any modification, for computing *Maximal* Independent Sets (mIS) that satisfy certain desired properties. While algorithms are known for the problem of finding a *Maximum* Independent Set (MIS), for which the desired property is the largest size of the set, to the best of our knowledge there are no algorithms known for the more general case for which we have a generic property. Currently, to the best of our knowledge, the only way to address this challenge is to list all mIS and select one of them. Thus, we argue that our method represents an advance for this combinatorial optimization problem. The reason for focusing the application on the best-shot public goods game in this paper is that it is an application of mIS that is well studied, and which allowed us to compare our method to well-established baselines to determine its validity and advantages.
>
> Furthermore, we also mentioned possible broader applications of our search plus the "proto-action" architecture in the context of other combinatorial optimization problems (which is also recommended to be brought forward by Reviewer cDVM). To expand, this represents a mechanism for learning a policy parametrization in a data-efficient way, since the number of actions is linearly dependant on the number of nodes in a graph. This is similar in spirit to [1], which considers embedding actions in a continous space, with two important differences: we propose to learn the embeddings end-to-end (rather than assuming them as given) and, furthermore, we adapt this for use with imitation, rather than reinforcement, learning -- which can be significantly more data-efficient where expert demonstrations can be provided. Anecdotally, based on our experimentation, parameterizing the policy in this way leads to an improved sample efficiency and generalizability boost. However, since we would need to present concrete, robust evidence and that the contribution is somewhat orthogonal to the network games focus of the present paper, we have not explicitly prioritized it.
>
> > My questions regarding the work lie in its significance to the NeurIPS community, but I hesitate somewhat to maintain that position (as it seems somewhat unfair).
>
> We appreciate that the reviewer does not adopt this position. We chose NeurIPS as the venue for our work for a series of reasons. The *computation of equilbria in multi-player games* and *graph combinatorial optimization problems*, broadly speaking, are areas of interest to the NeurIPS community. Furthermore, we think our manuscript is well-suited given our focus on a methodology based on machine learning. Past NeurIPS papers investigating the computation of equilbria in multi-player games include  [2, 3], whereas graph combinatorial optimization problems are the focus of [4, 5]. These are just some examples, but we believe it is possible to identify a variety of works in these research areas.
>
>
> ### References
>
> [1] Dulac-Arnold, G., Evans, R., van Hasselt, H., Sunehag, P., Lillicrap, T., Hunt, J., Mann, T., Weber, T., Degris, T. & Coppin, B. (2015). Deep reinforcement learning in large discrete action spaces. arXiv preprint arXiv:1512.07679.
>
> [2] McAleer, S., Lanier, J., Fox, R., & Baldi, P. (2020). Pipeline PSRO: A Scalable Approach for Finding Approximate Nash Equilibria in Large Games. NeurIPS 2020.
>
> [3] Farina, G., & Sandholm, T. (2020). Polynomial-time computation of optimal correlated equilibria in two-player extensive-form games with public chance moves and beyond. NeurIPS 2020.
>
> [4] Dai, H., Khalil, E. B., Zhang, Y., Dilkina, B., & Song, L. (2017). Learning combinatorial optimization algorithms over graphs. NeurIPS 2017.
>
> [5] Nazari, M., Oroojlooy, A., Snyder, L. V., & Takáč, M. (2018). Reinforcement learning for solving the vehicle routing problem. NeurIPS 2018.

---

> > ### Comment · Reviewer_ExWG · 2021-08-18
> > **Further comments**
> >
> > Thanks for your detailed and thoughtful replies.  Those are very helpful.
> >
> > I do think it is important to consider how to get society to follow the computed solution when computing the solution.  Computing the an efficient outcome is good, but if society can't be made to accept or follow it, it is not all that worthwhile.  Thus, as you mention, computing a set of solutions and then picking the one that is most likely to be implemented by society seems like a good way to go about the problem.  But I do agree that computing efficient solutions is a good first step.

---

### Decision · Program_Chairs · 2021-09-27

**Decision:**

Accept (Poster)

**Comment:**

The paper tackles graph-based public good games and proposes efficient search-based and learning-based algorithms for finding desirable equilibria. Based on all the reviews, responses, and discussions, here is the overall evaluation:
(+) It is a solid paper with a clearly defined problem (public good game on a graph) and analysis.
(+) It proposes tree-search-based (case-specific) and imitation learning-based (generalizable to new cases) algorithms for computing desirable pure strategy Nash equilibrium and these algorithms show superior performance in the experiments. The latter leverages the graph-neural network and is a new example of learning-powered combinatorial optimization with very good performance.
(-) The description of the algorithm, figure, and experimental results lack some details. The authors agree to provide additional results and details in their responses, which partially addresses the reviewers’ concerns.
(-) The team is not fully sure whether it can lead to a significant impact on society as it is hard to apply this algorithm to guide individuals in practice in a concrete application. Since the analysis of equilibrium is an important step towards a better understanding of public good games, the team still thinks the work is of value to the community.

Overall, the reviewer team has a positive view of the paper. We suggest the authors make changes they promised in the responses to improve the paper.